# Susceptibility of Synanthropic Rodents (*Mus musculus*, *Rattus norvegicus* and *Rattus rattus*) to H5N1 Subtype High Pathogenicity Avian Influenza Viruses

**DOI:** 10.3390/pathogens13090764

**Published:** 2024-09-05

**Authors:** Tatsufumi Usui, Yukiko Uno, Kazuyuki Tanaka, Tsutomu Tanikawa, Tsuyoshi Yamaguchi

**Affiliations:** 1Avian Zoonosis Research Center, Faculty of Agriculture, Tottori University, Tottori 680-0853, Japan; usutatsu@tottori-u.ac.jp (T.U.);; 2Technical Research Laboratory, IKARI SHODOKU Co., Ltd., Chiba 275-0024, Japan; kazuyuki-tanaka@ikari.co.jp (K.T.); tanikawa@ikari.co.jp (T.T.)

**Keywords:** house mice, black rats, brown rats, H5N1 subtype HPAIV

## Abstract

Synanthropic wild rodents associated with agricultural operations may represent a risk path for transmission of high pathogenicity avian influenza viruses (HPAIVs) from wild birds to poultry birds. However, their susceptibility to HPAIVs remains unclear. In the present study, house mice (*Mus musculus*), brown rats (*Rattus norvegicus*), and black rats (*Rattus rattus*) were experimentally exposed to H5N1 subtype HPAIVs to evaluate their vulnerability to infection. After intranasal inoculation with HA clade 2.2 and 2.3.2.1 H5N1 subtype HPAIVs, wild rodents did not show any clinical signs and survived for 10- and 12-day observation periods. Viruses were isolated from oral swabs for several days after inoculation, while little or no virus was detected in their feces or rectal swabs. In euthanized animals at 3 days post-inoculation, HPAIVs were primarily detected in respiratory tract tissues such as the nasal turbinates, trachea, and lungs. Serum HI antibodies were detected in HA clade 2.2 HPAIV-inoculated rodents. These results strongly suggest that synanthropic wild rodents are susceptible to infection of avian-origin H5N1 subtype HPAIVs and contribute to the virus ecosystem as replication-competent hosts. Detection of infectious viruses in oral swabs indicates that wild rodents exposed to HPAIVs could contaminate food, water, and the environment in poultry houses and play roles in the introduction and spread of HPAIVs in farms.

## 1. Introduction

Since 1997, H5 subtype high pathogenicity avian influenza (HPAI) viruses (HPAIVs) have repeatedly caused epidemics of HPAI and have caused massive damage to the poultry industry around the world. Wild aquatic birds belonging to the orders *Anseriformes* and *Charadriiformes*, which are the reservoir hosts of avian influenza viruses (AIVs), should be considered as potential original sources of HPAIV to poultry. However, direct contact with wild birds and poultry birds in poultry houses is unlikely to occur, especially in modern poultry systems that apply strict biosecurity measures. There could be intermediate factors in HPAIV transmission from wild birds to poultry. Potential vectors of HPAIV introduction may include feed, water, aviary equipment, the clothing and shoes of farm workers, flying insects, small birds, and mammals. Anything that could contact the feces or other excrement of HPAIV-carrying waterfowl should be considered a source or route of transmission of HPAIVs. To design more practical biosecurity measures in poultry operations, the route by which HPAIV is introduced to farms must be elucidated. In this regard, synanthropic animals that are frequently found around poultry farms, such as terrestrial wild birds and wild and domestic mammals, are some of the most potent risk factors [1,2,3].

Wild rodents, such as house mice (*Mus musculus*), brown rats (*Rattus norvegicus*), and black rats (*Rattus rattus*), may be the most prevalent mammals in the world’s poultry farms. Animals that are frequently observed on farms are more likely to be at risk than those that are rarely observed on farms [4]. Bacterial infections such as *Salmonella*, *Campylobacter*, and *Escherichia coli* are known as rodent-borne diseases on poultry farms. Wild rodents are also believed to play a role in the transmission of viruses such as AIVs, avian paramyxovirus 1, avian gammacoronavirus, and infectious bursal disease virus. Therefore, rodent control programs have been empirically and aggressively recommended as biosecurity measures for poultry operations to prevent the introduction and spread of rodent-mediated pathogens in farms. In the case of avian influenza, wild rodents are suspected to be HPAI vectors in poultry farms [4,5]. During the initial outbreak of H5N1 subtype HPAIV in Hong Kong in 1997, HI antibodies were detected in the sera of some brown rats captured in live poultry markets, although a virus was not isolated [6]. In 2015, an epidemiological study of house mice on farms experiencing HPAI outbreaks in Ohio, USA, did not find any virus-positive or antibody-positive cases [3]. However, in May 2024, H5N1 subtype HPAIV infections were reported in 66 house mice in New Mexico, USA, where H5N1 subtype HPAIV infections had occurred in dairy cattle (https://wahis.woah.org/#/in-review/4451 (accessed on 10 July 2024)). Although few in number, these cases of natural infection in brown rats and house mice imply that at least some synanthropic rodents are susceptible to HPAIV.

Wild house mice and brown rats that were experimentally inoculated with low pathogenicity AIVs were shown to be susceptible to avian-origin viruses and productively replicate viruses in their respiratory tract [7,8]. On the other hand, a few studies have shown a direct link between wild rodents and HPAIVs. In bank voles (*Myodes glareolus*), which are wild rodents sharing habitat with the wild reservoirs of AIVs, inoculation with avian-origin H5N1 and H7N1 subtype HPAIVs caused asymptomatic infection and resulted in the shedding of high amounts of virus [6,9]. In a black rat experimentally infected with H5N1 subtype HPAIV, a small amount of virus was isolated in the kidney and colon 3 days after infection [2]. In that study, the amount and duration of viral shedding from HPAIV-inoculated black rats after the challenge was not evaluated. Likewise, the susceptibility of house mice and brown rats to HPAIVs has not yet been explored by experimental inoculation.

In the past few years, many cases of HPAIV infection in mammals have been reported, raising concerns about the risk of transmission to humans [10]. The HPAIV susceptibility of rodents that live close to humans needs to be elucidated not only to protect poultry farms but also as a public health concern. In the present study, wild-caught house mice, brown rats, and black rats were experimentally inoculated with two H5N1 subtype HPAIVs, which had caused outbreaks in poultry and wild birds in the winters of 2007 and 2010, respectively, in Japan. They were observed to determine their susceptibility to HPAIVs as shown by clinical manifestations, viral shedding, and tissue distribution, as well as antibody response after challenge.

## 2. Materials and Methods

### 2.1. Animals (Wild and Laboratory Rodents)

Twenty-nine house mice (13 males and 16 females), 25 brown rats (13 males and 12 females), and 24 black rats (11 males and 13 females) captured and reared by Technical Research Laboratory, IKARI SHODOKU CO., LTD., in pest-control efforts were used in the present study. After transport to Tottori University, wild rodents were separately reared in custom-made animal cages (Natsume Seisakusho Co., Ltd., Tokyo, Japan). Because the body weights of most house mice ranged from 10 to 20 g, five house mice with less than 10 g body weight were separated as a small house mice group. Prior to virus challenge, blood was collected from all wild rodents using animal lancets (Medipoint, Inc., Mineola, NY, USA) under anesthesia. The absence of specific antibodies in their sera was confirmed by hemagglutination-inhibition (HI) tests against the challenge viruses, as described below.

For laboratory species of rodents, 22 four-week-old female BALB/c mice (*Mus musculus*) and 22 four-week-old female Sprague-Dawley (SD) rats (*Rattus norvegicus*) were purchased from Clea Japan, Inc., Tokyo, Japan.

### 2.2. Viruses

An HA clade 2.3.2.1 H5N1 subtype HPAIV, A/chicken/Shimane/1/2010 (Shima10), which was an index strain from the 2010–2011 poultry outbreaks in Japan [11], was kindly provided by the National Institute of Animal Health, Japan. An HA clade 2.2 H5N1 subtype HPAIV, A/mountain hawk-eagle/Kumamoto/1/2007 (Kuma07), was isolated from a mountain hawk-eagle (*Nisaetus nipalensis*) found dead on 4 January 2007 [12]. This isolate is genetically close to poultry isolates such as A/chicken/Miyazaki/K11/2007 (H5N1), which caused HPAI outbreaks in poultry on January and February 2007 in Japan [13]. The deduced amino acid sequence of the 2007 epidemic strains including Kuma07 have a lysine at amino acid 627 of the PB2 protein (PB2-627K), a mutation known to be associated with mammalian adaptation [14,15].

### 2.3. Inoculation of Rodents with HPAIVs

Eight each of house mice, black rats, brown rats, BALB/c mice, and SD rats were lightly anesthetized with 3 to 5% isoflurane gas inhalation and intranasally inoculated with 100 μL of 10^7^ fifty-percent egg infectious dose (EID_50_) of Shima10 or Kuma07 diluted in phosphate-buffered saline (PBS). The inoculum volume for the house mice experiment was reduced to 50 μL, containing 10^7^ EID_50_ of viruses. Three of the eight individuals of each species were euthanized at 3 days post-inoculation (dpi) (at 2 dpi for the house mice experiment) and the brain, nasal turbinate, trachea, lung, heart, spleen, liver, kidney, duodenum, rectum, and heparinized whole blood were collected for virus isolation. The remaining five animals were clinically observed for 12 days after virus challenge. Because of a scheduling problem, the observation period was 10 days for the SD rats inoculated with Shima10. Only for the BALB/c mice and SD rat experiments, another three animals as a control group were mock inoculated with 100 μL of PBS instead of viruses. Five small house mice with less than 10 g body weight were also challenged intranasally with 50 μL of 10^7^ EID_50_ of Kuma07 and clinically observed for 12 days.

To monitor virus shedding, oral and rectal swabs were collected from five individuals under anesthetized conditions at 1, 2, 3, 5, 7, and 12 dpi and suspended in 1 mL of PBS containing 10,000 unit/mL penicillin (Meiji Seika Pharma, Tokyo, Japan) and 10,000 μg/mL streptomycin (Meiji Seika Pharma, Tokyo, Japan). If available, fresh fecal samples of house mice and laboratory mice were collected instead of rectal swabs.

Tissue samples and fecal samples were homogenized in PBS containing antibiotics to obtain 20% suspensions. The swab, whole blood, fecal, and tissue samples were inoculated into 10-day-old embryonated chicken eggs (ECEs) (Aoki Breeder Farm, Tochigi, Japan) for calculation of EID_50_. At the end of the observation period, post-challenge sera were collected from all surviving animals and tested for specific antibodies against challenge viruses using an HI test to confirm infection. All experiments with infectious HPAIVs were performed in the biosafety level-3 containment laboratory at the Avian Zoonosis Research Center, Faculty of Agriculture, Tottori University.

### 2.4. Viral Shedding of Wild Rodents after Exposure to Shima10

To further evaluate the risk of wild rodents as potent vectors of HPAIVs, the amount and length of oral shedding in wild rodents were examined after dose-dependent exposure to Shima10 HPAIV. In the case of house mice, 8 animals were divided into two groups of four mice and intranasally inoculated with 10^5^ and 10^3^ EID_50_ of Shima10. For brown rats, 9 animals were divided into three groups of 3 rats and intranasally inoculated with 10^5^, 10^3^, and 10^1^ EID_50_ of Shima10. Eight black rats were intranasally inoculated with 10^5^ (*n* = 3), 10^3^ (*n* = 2), and 10^1^ (*n* = 3) EID_50_ of Shima10. These animals were monitored for viral shedding in oral swabs for 7 days after challenge.

### 2.5. Hemagglutination-Inhibition (HI) Test Against the Challenge Viruses

The pre- and post-serum samples were tested for HI antibodies according to the standard method (WHO Manual on Animal Influenza Diagnosis and Surveillance, http://www.who.int/iris/handle/10665/68026 (accessed on 9 May 2020)), with slight modifications. For this assay, a solution of receptor-destroying enzyme II (Denka Seiken Co., Niigata, Japan) was freshly prepared by dissolving 2 mg lyophilized enzyme in 20 mL sterile saline. For each assay, 75 μL of enzyme solution was mixed with 25 μL of serum. The mixture was incubated at 37 °C for 18 h and heat-inactivated at 56 °C for 30 min. Viruses such as hemagglutinin (HA) antigen were adjusted to 4 HA units/25 μL, and then the HI titer was measured with 0.5% packed chicken red blood cells. The lower limit of detection was 1:4 for this HI test (i.e., values less than 4 HI were undetectable), except for the SD rats after Kuma07 exposure, for whose serum the lower limit was 1:10.

## 3. Results

### 3.1. Susceptibility of Rodents to H5N1 Subtype HPAIVs

#### 3.1.1. House Mice

House mice whose body weights ranged from 10.1 to 13.0 g (11.9 ± 1.1 g; *n* = 5) and from 11.3 to 16.5 g (13.1 ± 2.0 g; *n* = 4) were intranasally inoculated with Shima10 and Kuma07 HPAIVs. After challenge, all house mice survived for 12 days without notable clinical symptoms, except for one of the five Kuma07-inoculated house mice that accidentally died during intranasal inoculation. While the Shima10-inoculated house mice slightly gained body weight during the observation period, one of the four Kuma07-inoculated house mice lost as much as 25% of its body weight by 10 dpi but slightly gained body weight at 12 dpi (Figure 1A).

Low titers of Shima10 were detected in oral swabs from two out of five house mice by 3 dpi (Figure 2A), whereas HPAIVs were not detected in fecal samples. Shima10 virus was isolated from the nasal turbinates (10^3.7 ± 0.5^ EID_50_/g) and lung (10^4.2 ± 0.0^ EID_50_/g) of the three house mice euthanized at 2 dpi, and two of them showed viremia (Table 1). Four house mice inoculated with Kuma07 orally shed the virus for a maximum of 3 days (Figure 2B). One house mouse showing 25% loss of body weight by 10 dpi orally shed the virus at the highest titers, 10^2.75^ and 10^2.25^ EID_50_/mL, at 1 and 2 dpi. Extremely low titers of Kuma07 virus (10^1.75^ EID_50_/g) were detected in the feces of three house mice at 1 or 2 dpi. At 2 dpi, Kuma07 was isolated from the nasal turbinates (10^3.6 ± 1.2^ EID_50_/g) of all three mice and the trachea (10^1.9^ and 10^2.4^ EID_50_/g) of two mice but not from the lung or blood (Table 1).

Specific HI antibodies were detected in the sera of all four Kuma07-inoculated house mice at 12 dpi, while Shima10-inoculated house mice remained seronegative (Table 2).

#### 3.1.2. Small House Mice (HA Clade 2.2 H5N1 Subtype HPAIV)

When five house mice, ranging from 6.3 to 9.0 g (7.4 ± 1.1 g), were challenged with Kuma07 HPAIV, three of them suddenly died at 3 dpi without changes in appearance. One of the remaining house mice also died at 8 dpi with 25% body weight loss (Figure 1B). The surviving house mouse showed no weight loss nor clinical signs during the observation period. At 1 and 2 dpi, Kuma07 was isolated from oral swabs from all five of the mice (Figure 2C), but no virus was detected in feces. Postmortem examination showed severe hemorrhage of the lung in three of the dead mice. Kuma07 was detected in the lungs (10^5.1 ± 0.1^ EID_50_/g) and hearts (10^3.6 ± 0.2^ EID_50_/g) of 3 of the mice that died at 3 dpi (Table 1) and in the lung (10^3.2^ EID_50_/g) and heart (10^1.4^ EID_50_/g) of the mouse that died at 8 dpi. The surviving mouse shed virus at 1, 2, and 5 dpi and then seroconverted (≥256 HI) by 12 dpi (Table 2).

#### 3.1.3. Brown Rats

At the time of inoculation, the body weights of brown rats ranged from 159 to 184 g (168.8 ± 9.1 g; *n* = 5) for the Shima10 experiment and from 149 to 230 g (204.8 ± 29.2 g; *n* = 5) for the Kuma07 experiment. In both groups, none of the rats showed any clinical signs after challenge, and slightly gained body weight during the experiment (Figure 1C).

Shima10 HPAIVs were detected in oral swabs of 4 to 5 of the rats for days 1-3 but not after 5 dpi (Figure 2D). At 2 dpi, oral shedding of Shima10 was detected in all five rats at an average titer of 10^1.45 ± 0.51^ EID_50_/mL. At 3 dpi, Shima10 was detected in the nasal turbinates (10^2.7 ± 0.5^ EID_50_/g) of all three rats and the lung (10^2.2^ EID_50_/g) of one rat (Table 1). The detection of Kuma07 HPAIV in oral swabs peaked at 1 dpi (10^2.2 ± 0.7^ EID_50_/mL), and the highest titer reached 10^3.25^ EID_50_/mL in one rat. Kuma07 was detected in this rat for a maximum of 5 days (Figure 2E). Kuma07 was broadly recovered from the respiratory tract, including the nasal turbinates (10^1.4^ and 10^4.4^ EID_50_/g), trachea (10^3.9^ and 10^1.7^ EID_50_/g), and lung (10^2.9^ and 10^2.4^ EID_50_/g) of at least two of the three rats. All rectal swabs from both groups were negative for both of the viruses. Seroconversion by 12 dpi was detected in all Kuma07-inoculated rats but none of the Shima10-inoculated rats (Table 2).

#### 3.1.4. Black Rats

Body weights of the black rats ranged from 97 to 124 g (107.4 ± 9.1 g; *n* = 5) for the Shima10 experiment and from 99 to 125 g (111.2 ± 11.6 g; *n* = 5) for the Kuma07 experiment. After virus inoculation, all of the rats survived for 12 days without exhibiting any clinical signs or losing body weight (Figure 1D).

In both groups, viral shedding to the oral cavity lasted for 5 days after inoculation. Shima10 viruses were detected in oral swabs of all five rats at 2 dpi (10^1.86 ± 0.78^ EID_50_/mL) and 3 dpi (10^1.85 ± 0.58^ EID_50_/mL) (Figure 2F). The titer of Shima10 in the oral swab of one rat exceeded 10^3^ EID_50_/mL at 3 and 5 dpi. At 3 dpi, Shima10 was isolated from the nasal turbinates (10^3.5 ± 0.8^ EID_50_/g) of all three rats and detected in the lung (10^1.4^ and 10^1.4^ EID_50_/g) of two rats (Table 1). Oral shedding of the Kuma07 virus started in all rats by 2 dpi and continued for a maximum of 5 days (Figure 2G). The average titer of Kuma07 exceeded 10^2^ EID_50_/mL at 1 and 2 dpi. Kuma07 was recovered from the lungs (10^4.0 ± 0.7^ EID_50_/g) of all three rats. On the other hand, no virus was detected in the rectal swabs from any of the rats inoculated with Shima10 or Kuma07 HPAIVs.

Four of the five of Kuma07-inoculated rats seroconverted by 12 dpi. No HI antibodies were detected in the serum of a black rat that shed a low titer of virus and only at 1 and 2 dpi (10^1.75^ and 10^0.75^ EID_50_/mL). None of the rats inoculated with Shima10 produced detectable HI antibodies at 12 dpi (Table 2).

### 3.2. HPAIV Infection in BALB/c Mice (M. musculus) and SD Rats (R. norvegicus)

To compare the HPAIV susceptibilities among wild rodents, BALB/c mice and SD rats, which are laboratory species of house mice and brown rats, respectively, were intranasally inoculated with Shima10 (*n* = 5) and Kuma07 (*n* = 5). For both H5N1 strains, the mice displayed severe clinical signs, such as lethargy and ruffled fur, and significantly lost weight after 2 dpi (Figure 1E). All five BALB/c mice inoculated with Shima10 died at 6 to 8 dpi, and all five inoculated with Kuma07 died at 4 to 5 dpi. Oral shedding of both viruses was detected on the first day after inoculation. Shima10 peaked at 1 dpi (10^2.36 ± 0.49^ EID_50_/mL), and viral shedding was detected for up to 7 days (Figure 2H). The oral titer of Kuma07 gradually increased and peaked at 3 dpi (10^2.80 ± 0.37^ EID_50_/mL) (Figure 2I). Kuma07 virus was detected in the feces of two of the five mice at 1 dpi, although these titers were extremely low (10^1.75^ EID_50_/g). After 2 dpi, fecal samples were not collected from either group of mice because of a reduction or disappearance of defecation. Both HPAIVs were isolated from the peripheral blood and multiple organs of BALB/c mice (*n* = 3 each) at 3 dpi (Table 1).

In contrast to the BALB/c mice, HPAIV-inoculated SD rats showed no clinical signs and steadily gained body weight throughout the experiment (Figure 1F). Oral shedding at low virus titer started in all Shima10-inoculated rats by 3 dpi and lasted until 5 dpi (Figure 2J). Shima10 was isolated from the nasal turbinates and lung of all three rats at 3 dpi (Table 1). Oral shedding of the virus was detected in all five Kuma07-inoculated rats at 1 dpi (10^2.30 ± 0.29^ EID_50_/mL) and lasted for up to 3 days in rats (Figure 2K). At 3 dpi, Kuma07 was detected in the lung (10^1.9^ EID_50_/g) from one of the three rats (Table 1). Rectal swabs were all negative for Shima10 and Kuma07. All SD rats inoculated with Kuma07 seroconverted by 12 dpi, but like the wild rodents, none of the Shima10-inoculated rats produced detectable HI antibodies at 10 dpi (Table 2).

### 3.3. Effect of Shima10 HPAIV Dose on Oral Shedding in Wild Rodents

To further evaluate the risk of wild rodents as vector animals, the amount and length of virus shedding from wild rodents exposed to 10^5^, 10^3^, and 10^1^ EID_50_ of Shima10 HPAIV were examined. Shima10 was detected in the oral swabs in two out of the four house mice inoculated with 10^5^ and 10^3^ EID_50_ (Figure 3A), indicating that house mice could shed detectable levels of the virus after exposure to only 10^3^ EID_50_ of Shima10. Oral shedding was detected in all three of the brown rats in the 10^5^ EID_50_ group and in one brown rat of 10^3^ EID_50_ group (Figure 3B). In a brown rat inoculated with 10^5^ EID_50_ of Shima10, the titer of the virus reached 10^3.25^ EID_50_/mL at 2 dpi. In black rats, oral shedding was detected only in the 10^5^ EID_50_ group, but all three black rats steadily shed the virus by 5 dpi (Figure 3C).

## 4. Discussion

Wild house mice, brown rats, and black rats, which are abundant in most poultry farms, are thought to have been a contributing factor to HPAI outbreaks across the world [4,5]. However, little is known about their HPAIV susceptibility, clinical signs, route of viral shedding, or duration of shedding. To the best of our knowledge, the only rodents that have been experimentally infected with HPAIV are black rats and bank voles. One study [2] showed black rats have low susceptibility to various H5N1 subtype HPAIVs, since little or no viruses were recovered at 3 dpi. Another study [9] showed that bank voles, which are common wild mice throughout Europe and Asia, were infected asymptomatically with H5N1 and H7N1 subtype HPAIVs and shed viruses in their nasal washes. In the present study, we infected a larger variety of rodents with two H5N1 subtype HPAIVs. Shima10 belongs to HA clade 2.3.2.1 HPAIVs, which caused a nationwide epidemic among poultry and wild birds in Japan in the 2010–2011 season [11]. Kuma07 belongs to HA clade 2.2 HPAIVs (formerly called Qinghai Lake lineage), which were isolated from a raptor and poultry in Japan during the 2006–2007 season [12]. The present results strongly suggest that synanthropic wild rodents are susceptible to infection of avian-origin H5N1 subtype HPAIVs. Detection of infectious virus in the oral cavity indicates that wild rodents exposed to HPAIVs shed virus through saliva and could contaminate food, water, and equipment in the poultry house.

Among the three rodent species examined in this study, black rats appeared to shed the most virus and for the longest duration (Figure 2) and thus were assumed to have the most potential to transmit HPAIVs to poultry. Hiono et al. (2016) infected four black rats with HA clade 2.3.2.1 virus but detected the virus in only one of them, and in that rat, the virus loads were extremely low and confined to the kidney and intestine [2]. In the present study, our finding of HA clade 2.3.2.1 or clade 2.2 viruses in the nasal turbinates or lungs, but not the kidney or intestine, of black rats (Table 1) suggests that the viruses replicates in respiratory tissues, although it was unclear which cells were infected. The analysis of oral swabs collected periodically revealed that black rats shed viruses for several days after inoculation of both HPAIVs. Stable viral shedding was observed in black rats even after exposure to smaller doses (10^5^ EID_50_) of Shima10 (Figure 3C). Although the virus strains and the amount of virus inoculum differed from the strains and amount used in the previous study [2], the present results indicate that black rats are more susceptible to HPAIVs than previously reported [2]. The differences of viral shedding among species or individuals are likely partly due to differences in innate immune responses to H5N1 subtype HPAIVs.

As for the Shima10 virus, a fifty percent chicken infectious dose (CID_50_) for 4-week-old specific pathogen-free (SPF) chickens was calculated as 10^3^ EID_50_ [11], which was detected in oral swabs from one black rat (Figure 2F). Although the CID_50_ of Kuma07 was not defined, that of an HA clade 2.2 H5N1 virus (A/chicken/Miyazaki/K11/2007), which is genetically close to Kuma07, was shown to be 10^2.5^ EID_50_ using 4-week-old SPF chickens [13]. In the Kuma07 experiments, two black rats, two brown rats, and one house mouse discharged virus exceeding the deemed CID_50_ (10^2.5^ EID_50_) at 1 dpi (Figure 2B,E,G). A high titer of virus shedding (10^3^ EID_50_) was also detected in one of the black rats at 2 dpi. These results suggest that some wild rodents, notably black rats, exposed to HPAIV shed enough virus from the oral cavity to infect chickens.

While wild mice and rats stayed active in their cages after HPAIV challenge, laboratory BALB/c mice showed marked weight loss and changes in appearance, such as ruffled fur and hunched posture, and finally died. This noticeable difference between biologically identical species (*Mus musculus*) could be explained in part by the finding that the immune systems of wild mice are more active than those of laboratory bred mice [16]. Although a high titer of Kuma07 virus exceeding the assumed CID_50_ (10^2.5^ EID_50_) was detected in oral swabs of BALB/c mice at 3 dpi (Figure 2I), these mice did not really contaminate the ambient surroundings because they had laid still from 2 dpi to death at 4-5 dpi. If virus carriers were too sick to enter and run around poultry houses or died outside the poultry house, the risk of transmission and spread of viruses would be low. The lack of clinical manifestations in SD rats appears to resemble those of black rats and other wild rodents, although the length and extent of viral shedding were slightly less compared to black rats (Figure 2J,K). Bank voles were also shown to be infected asymptomatically with HPAIV and shed viruses in their nasal washes [9]. From these findings, based on the challenge experiments, it was shown that wild rodents are subclinically infected with HPAIVs. This means that wild synanthropic rodents after HPAIV exposure can pollute the environment of poultry premises by roaming around while shedding infectious viruses and thus play a role transmitting or spreading viruses.

Even though wild rodents can be infected subclinically with HPAIVs, there are exceptions, like the small house mice that died suddenly after Kuma07 HPAIV infection. The wild house mice had greater interindividual variation in their immune ability [16]. Immune status may be compromised by other pathogens or stress from the environment. Thus, there are cases in which wild animals, especially immature or compromised animals, die after HPAIV infection. Immunity in wildlife is still largely unexplored. But a better understanding of the viral susceptibility of wild animals requires analysis of innate and acquired immunity. Postmortem observations revealed that Kuma07 HPAIVs highly replicated in the lungs and hearts of the dead house mice (Table 1), indicating that the viruses might be transmitted to poultry through pecking at the rodent carcasses, in which viable viruses remained.

More recently, natural infection of house mice with HA clade 2.3.4.4b H5N1 subtype HPAIV was reported to the World Organization for Animal Health (WOAH) in June 2024 (https://wahis.woah.org/#/in-review/4451 (accessed on 10 July 2024)). The route of infection has not been elucidated, but it is assumed that the infection was caused by ingestion of unpasteurized milk from HPAIV-infected dairy cattle [17]. It was previously reported that influenza virus nucleic acids were detected in Norway rats (brown rat) inhabiting a city area [18]. Both cases imply that the source of infection is not related to waterfowl reservoirs. However, these findings provided a direct connection between wild rodents and influenza viruses. If wild rodents are infected with HPAIVs by association with waterfowl in ponds or puddles around poultry farms, they will remain active and will enter and contaminate poultry houses with HPAIVs through their saliva or respiratory excretions. That is one way in which HPAIVs can be transmitted from wild rodents to poultry. Alternatively, wild rodents may also spread HPAIVs between poultry houses. In this case, wild rodents might be infected with HPAIVs, or they might pick up the virus on their fur or palms in poultry houses containing infected poultry. Rodents carrying HPAIV may also be a source of infection for predators such as weasels and martens, which target rodents and invade poultry houses [19].

In the present study, no HI antibodies were detected in the sera of any of the Shima10-inoculated group (Table 2). In some studies in which wild rodents were experimentally infected with avian-origin viruses, antibodies to the viruses were not detected. These included a black rat infected with HA clade 2.3.4.4 H5N1 subtype HPAIV [2] and Norway rats infected with LPAIVs [7]. These viruses were used in inoculation trials prior to adaptation to mammals. Kuma07 has a lysine residue at position 627 in PB2 (627K), which reportedly contributes to the adaptation to mammals [14,15,20], while Shima10 has a glutamate at this position (627E). The reason for the lack of antibody response in Shima10-inoculated groups remains unclear. However, Shima10 was less adapted to mammals than Kuma07. Shima10 and Kuma07 do not harbor the HA mutations Q226L and G228S, which have been implicated in increasing the affinity to human-type α2,6 receptor [21]. If Shima10 acquires these adaptive mutations after infecting rodents, it may change enough to induce antibody production. Future experimental infections of rodents with HPAIVs should monitor the well-known PB2 mutation (E627K) and HA mutations (Q226L and G228S) to estimate the potential adaptive capacity of the viruses.

Demonstrating viral infection at the tissue level requires genetic detection, virus isolation, or immunohistochemical staining [22]. A weakness of this study was that Shima10 infection was not confirmed by histopathological examinations or detection of an antibody response. However, our results strongly suggest that both Shima10 and Kuma07 can replicate in the respiratory tract of the synanthropic wild rodents. Future studies of susceptibility to HPAIV in wild rodents and other mammals should use a variety of epidemic strains, especially HA clade 2.3.4.4b lineage viruses that have been circulating in recent years [10].

## 5. Conclusions

The present study examined the susceptibilities of common rodents that inhabit poultry farms, including house mice (*Mus musculus*), brown rats (*Rattus norvegicus*), and black rats (*Rattus rattus*) to H5N1 subtype HPAIVs by experimental infection. Our results revealed that synanthropic wild rodents can be infected with HPAIVs in a subclinical manner and were able to actively shed the virus from the oral cavity for 3–5 days after HPAIV exposure. The primary site of HPAIV multiplication was the respiratory tract, with little or no discharge from the gastrointestinal tract. Because the course of HPAIV infection typically varies among virus strains, laboratory investigations using a variety of virus strains, especially those prevalent in recent years, are required. Replication-competent hosts in poultry farms definitely present a high risk in terms of avian flu control and public health concerns. Further rodent control measures are required for biosecurity. While aiming to eliminate wild rodents from poultry farms, hygiene management must be taken to prevent the spread of the disease to other animals and to protect humans from HPAIV infection.

## Figures and Tables

**Figure 1 pathogens-13-00764-f001:**
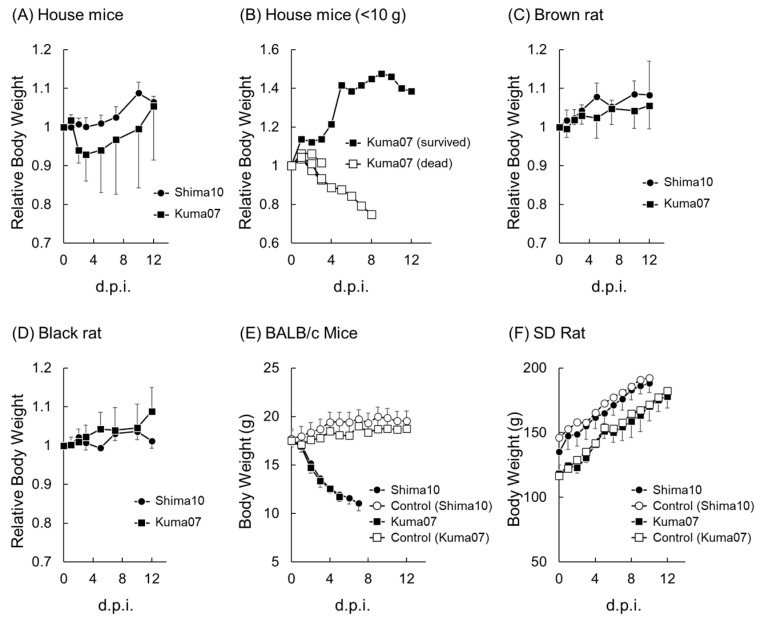
Changes in body weight of house mice (**A**,**B**), brown rats (**C**), black rats (**D**), BALB/c mice (**E**) and SD rats (**F**) inoculated with HPAIVs. In all panels except (**B**), error bars indicate standard deviation. d.p.i., days post inoculation.

**Figure 2 pathogens-13-00764-f002:**
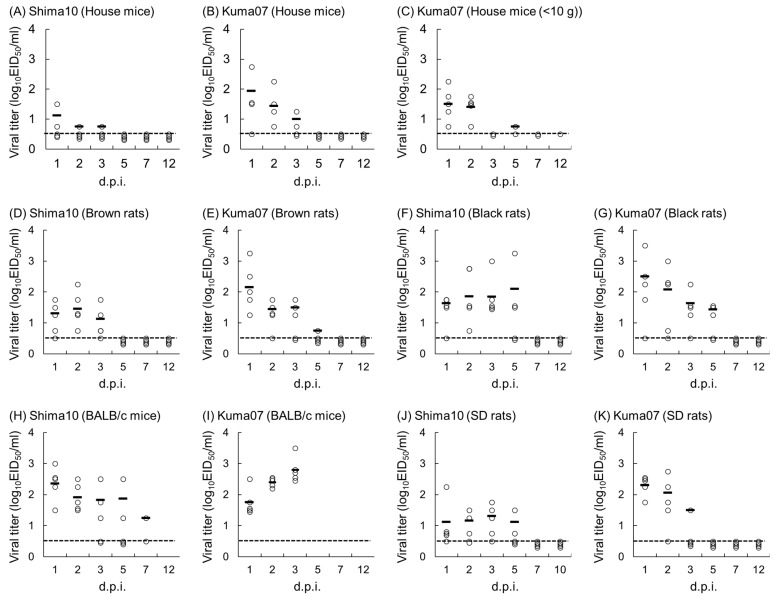
Changes in oral shedding of HPAIVs in wild and laboratory rodents. House mice (**A**–**C**), brown rats (**D**,**E**), black rats (**F**,**G**), BALB/c mice (**H**,**I**), and SD rats (**J**,**K**) were intranasally inoculated with Shima10 (**A**,**D**,**F**,**H**,**J**) or Kuma07 (**B**,**C**,**E**,**G**,**I**,**K**) HPAIVs. White circles represent the viral titers detected in the oral swab for each individual, and horizontal lines indicate the mean titer of positive samples. Dashed lines indicate the detection limit of the viruses (10^0.5^ EID_50_/mL). d.p.i., days post inoculation.

**Figure 3 pathogens-13-00764-f003:**
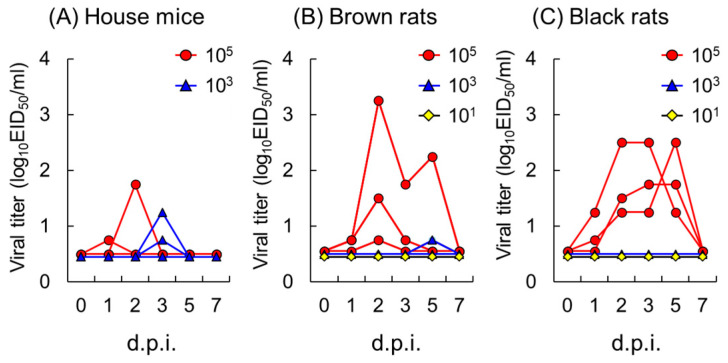
Changes in oral shedding of HPAIV in wild rodents inoculated with different titers of Shima10. House mice (**A**), brown rats (**B**), and black rats (**C**) were intranasally inoculated with 10^5^, 10^3^, or 10^1^ EID_50_ of Shima10 HPAIV. The amounts of virus in the oral swabs per individual are shown with red circles (10^5^ EID_50_ group), blue triangles (10^3^ EID_50_ group), and yellow diamonds (10^1^ EID_50_ group). d.p.i., days post inoculation.

**Table 1 pathogens-13-00764-t001:** Virus recovery from HPAIV-inoculated rodents at 3 dpi.

		Number of Animals from Which Virus Was Recovered at 3 dpi (Average Virus Titer (log_10_ EID_50_/g)) ^a^										
Animals ^b^	Virus	Peripheral Blood ^c^	Brain	Nasal Turbinate	Trachea	Lung	Heart	Spleen	Liver	Kidney	Duodenum	Rectum
House mice	Shima10	2 (2.0)	0	3 (3.7)	1 (1.4)	3 (4.2)	1 (2.4)	0	0	0	0	0
	Kuma07	0	0	3 (3.6)	2 (2.2)	0	0	0	0	0	0	0
(<10 g)	Kuma07 ^d^	n/a	0	n/a	n/a	3 (5.1)	3 (3.6)	0	1 (1.9)	1 (1.9)	0	1 (1.4)
Brown rats	Shima10	0	0	3 (2.7)	0	1 (2.2)	0	0	0	0	0	0
	Kuma07	0	0	2 (2.9)	2 (2.8)	2 (2.7)	0	0	0	0	0	0
Black rats	Shima10	0	0	3 (3.5)	0	2 (1.4)	0	0	0	0	0	0
	Kuma07	0	0	n/a	n/a	3 (4.0)	0	0	0	0	0	0
BALB/c mice	Shima10	3 (3.8)	1 (1.9)	3 (3.6)	3 (3.0)	3 (6.1)	3 (2.3)	3 (2.7)	1 (1.4)	1 (1.9)	0	1 (1.4)
	Kuma07	2 (4.5) ^e^	3 (3.3)	n/a	n/a	3 (7.7)	3 (4.5)	3 (3.1)	3 (4.3)	3 (3.5)	1 (1.4)	1 (2.2)
SD rats	Shima10	0	0	3 (3.9)	2 (2.2)	3 (2.5)	0	1 (1.4)	0	0	0	0
	Kuma07	0	0	n/a	n/a	1 (1.9)	0	0	0	0	0	0

^a^ Virus titers are represented as the geometric mean value based on positive samples. Detection limit is <1.2 log_10_ EID_50_/g. ^b^ Each of the three animals was inoculated with HPAIV. ^c^ Blood viral titers are represented as log_10_ EID_50_/mL. Detection limit of blood viral titer is <1.5 log_10_ EID_50_/mL. ^d^ The three house mice that died at 3 dpi were necropsied. ^e^ Blood was collected from two of the three Kuma07-inoculated BALB/c mice at 3 dpi because one died before sampling. n/a, not applicable (no sample).

**Table 2 pathogens-13-00764-t002:** HI titers in sera of rodents inoculated with HPAIVs at 12 dpi.

HPAIV	House Mice	House Mice (<10 g)	Brown Rats	Black Rats	BALB/c Mice	SD Rats ^a^
Kuma07	≥256	≥256	64	128	dead	≥640
	128	dead	32	64	dead	320
	128	dead	32	64	dead	320
	64	dead	8	64	dead	320
	(dead)	dead	8	<4	dead	80
Shima10	<4	n/a	<4	<4	dead	<4
	<4		<4	<4	dead	<4
	<4		<4	<4	dead	<4
	<4		<4	<4	dead	<4
	<4		<4	<4	dead	<4

^a^ Sera from Shima10-inoculated SD rats were collected at 10 dpi. n/a, not applicable.

## Data Availability

The original contributions presented in the study are included in the article; further inquiries can be directed to the corresponding author.

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
