# Peer review of "Susceptibility of Synanthropic Rodents (Mus musculus, Rattus norvegicus and Rattus rattus) to H5N1 Subtype High Pathogenicity Avian Influenza Viruses"

_pathogens, 2024, doi:10.3390/pathogens13090764_

Round 1

Reviewer 1 Report

Comments and Suggestions for Authors

Comments on the Quality of English Language

Needs to improve

Reviewer 2 Report

Comments and Suggestions for Authors

-Did the study investigate the presence of any specific mutations in the H5N1 strains used that are known to influence virulence or host range? If so, what were these mutations?

-How does the study address the existing gaps in the literature regarding the role of synanthropic rodents in the transmission of H5N1 subtype HPAIV?

-Could you elaborate on the specific ethical guidelines and approvals followed for the use of animals in this study?

- Can you explain the rationale behind the observation periods chosen for the different rodent species?

-How do you account for the variations in viral shedding observed among the different rodent species?

-How do your findings on the susceptibility of synanthropic rodents to H5N1 compare with previous studies on other rodent species or mammals?

-What biological or ecological factors might explain the different levels of susceptibility and viral shedding observed among the rodent species studied?

-How might the findings of this study influence biosecurity measures on poultry farms, specifically in regions with high rodent populations?

-What are the key limitations of your study, and how do they impact the generalizability of the findings?

-What specific future research directions do you recommend to further explore the role of synanthropic rodents in HPAIV transmission?

-Was there any evidence of interspecies transmission of H5N1 among the rodents during the study period?

-Can you provide more details on the histopathological findings in the tissues of infected rodents? How do these findings correlate with the observed clinical signs?

-How do the results of this study contribute to our understanding of zoonotic transmission of HPAIV to humans?

-Did the study find any mutations associated with increased viral replication or shedding in specific rodent species?

-Based on the mutations observed, what recommendations can be made for monitoring H5N1 strains in wild rodent populations? 

-How do the findings regarding mutations contribute to the broader understanding of HPAIV evolution and adaptation in non-avian hosts?

Comments on the Quality of English Language

Extensive editing of English language required

Reviewer 3 Report

Comments and Suggestions for Authors

The authors evaluated susceptibility of wild rodents to H5N1 HPAIVs by experimental infection. The authors found seroconversion after challenge in wild animals suggested these animals can be infected by H5N1 HPAIV but virus shedding was minimal. The authors have made a good attempt at adding value to the discussion of source of HPAIV transmission to poultry farms. Specific comments follow.

Major points:

1.     Line 82: Please add ethics statement for the animal experiments.

2.     Line 92: Please add HI assay method or reference of it.

3.     Please consider to add a following reference and to discuss.

N Engl J Med. 2024 Jul 4;391(1):87-90. doi: 10.1056/NEJMc2405495. Epub 2024 May 24. Cow's Milk Containing Avian Influenza A(H5N1) Virus - Heat Inactivation and Infectivity in Mice

Minor points:

1.     Line 267: Please use superscript or subscript in “105 and 103 EID50”.

2.     Line 377: Please add a reference for sequences.

Round 2

Reviewer 1 Report

Comments and Suggestions for Authors

Ok

Author Response

Thank you for taking your valuable time to check our manuscript again.

In this revision, we have revised the description of the manuscript, focusing on Discussion to make it clearer. In addition, some references on histopathological analysis and adaptive mutations have been added. All changes in the revised manuscript have been highlighted in red-colored text. The English text was reviewed by a native English speaker again.

Reviewer 2 Report

Comments and Suggestions for Authors

Thank you for addressing several aspects of the study in your responses. To further enhance the manuscript and address all concerns comprehensively, we request additional clarification on the following technical points:

  1. What specific reasons prevented the performance of histopathological analysis on infected rodent tissues, and what alternative methods were employed to assess the infection's impact on these tissues?
  2. Which tissues were collected for potential histopathological examination, and could you provide a detailed rationale for selecting these specific tissues?
  3. Given the limitations in obtaining tissue samples, what strategies could be implemented in future studies to ensure comprehensive histopathological analysis of H5N1 infection in rodents?
  4. Were any qualitative observations or preliminary assessments made regarding the histopathological effects of infection, and if so, what were the findings?
  5. Given that no genetic analysis of mutations was performed in this study, how might the known mutation E627K in PB2, associated with mammalian adaptation, influence the observed results in the rodent models used?
  6. Can you describe the potential impact of mutations in the HA or PB2 proteins on viral replication and transmission in rodents, based on existing literature or theoretical models?
  7. Were there any observable trends in viral shedding or replication that might suggest the presence of adaptive mutations, even though genetic analysis was not conducted?
  8. What methodologies would you recommend for incorporating genetic analysis of H5N1 strains in future studies to investigate mutation-related adaptations in rodent models?
  9. How does the absence of genetic analysis impact the interpretation of your study's findings with respect to HPAIV evolution and adaptation in non-avian hosts?
  10. What specific genetic markers or mutations should be monitored in future research to better understand the adaptation of H5N1 viruses in rodent populations?
Comments on the Quality of English Language

Moderate editing of English language required.
